# Exploring the Multifaceted Neuroprotective Mechanisms of Bovine Lactoferrin in a Cell Culture Model of Parkinson’s Disease

**DOI:** 10.3390/ijms262311312

**Published:** 2025-11-22

**Authors:** Giusi Ianiro, Noemi Martella, Antonella Niro, Mayra Colardo, Piera Valenti, Giovanni Musci, Antimo Cutone, Marco Segatto

**Affiliations:** 1Department of Biosciences and Territory, University of Molise, 86090 Pesche, Italy; giusy.ianiro@unimol.it (G.I.); noemi.martella@unimol.it (N.M.); a.niro2@studenti.unimol.it (A.N.); mayra.colardo@unimol.it (M.C.); giovanni.musci@unimol.it (G.M.); marco.segatto@unimol.it (M.S.); 2Department of Public Health and Infectious Diseases, Sapienza University of Rome, 00185 Rome, Italy

**Keywords:** Parkinson’s disease, lactoferrin, rotenone, neuroprotection, oxidative stress, iron homeostasis

## Abstract

Parkinson’s disease (PD), the second most common neurodegenerative disease, is characterized by the progressive degeneration of dopaminergic neurons in the substantia nigra pars compacta along with the aggregation of α-synuclein in Lewy bodies. Among the pathological mechanisms involved is the alteration of iron homeostasis, which promotes oxidative stress and neuronal damage. Despite therapeutic advances, today, no treatment is available to modify the course of the disease. In this study, we investigated for the first time the neuroprotective potential of bovine lactoferrin (bLf) in both its Native (Nat-) and Holo forms, using rotenone-treated N1E-115 cells to mimic PD phenotype. The results showed that the Nat-bLf was more effective than Holo-bLf in counteracting rotenone-induced cytotoxicity and neurite retraction, preserving neuronal morphology and promoting neuritogenesis, as evidenced by increased β3-Tubulin and Growth-Associated Protein-43 markers (GAP-43). Both forms of bLf preserved Tyrosine Hydroxylase (TH) levels, crucial for dopamine synthesis, reduced the DNA damage marker γ-H2Ax and prevented rotenone-induced downregulation of Divalent Metal Transporter-1 (DMT-1) and Ferroportin (Fpn), key proteins involved in iron uptake and release, thereby limiting intracellular iron accumulation. Notably, only Nat-bLf reduced the levels of α-synuclein and markers of oxidative damage. Conversely, Holo-bLf exhibited pro-oxidant effects and increased α-synuclein accumulation even in absence of rotenone. Overall, these results highlight the differential neuroprotective effects of both Nat- and Holo-form, resulting from their distinct iron saturation level and their ability to modulate protein expression, with the native form emerging as a promising candidate for therapeutic strategies to counteract PD-associated neurodegeneration.

## 1. Introduction

Parkinson’s disease (PD) is a progressive neurodegenerative disorder affecting approximately 2–3% of people over the age of 65, making it the second most prevalent neurodegenerative condition after Alzheimer’s disease [1]. Clinically, PD shows a combination of motor symptoms (tremors, bradykinesia, rigidity and postural instability) and non-motor symptoms (depression and cognitive decline) [2]. Despite decades of research, no resolutive therapies are available, and current treatments are only aimed at alleviating symptomatology. The pathological features of PD are mostly caused by the loss of dopaminergic neurons in the Substantia Nigra pars compacta (SNpc), which is accompanied by the intracellular accumulation of misfolded α-synuclein in Lewy bodies, as well as pronounced oxidative stress and neuroinflammation [3]. A crucial factor contributing to PD pathology is excessive iron accumulation observed in the SNpc of affected patients [4], leading to the production of hydroxyl radicals via Fenton reaction and redox-driven neurotoxicity. Redox stress, together with mitochondrial dysfunction, are key drivers of α-synuclein aggregation. Iron metabolism is tightly regulated in the Central Nervous System (CNS), with the transporter Divalent Metal Transporter-1 (DMT-1) playing a pivotal role in the uptake of non-transferrin-bound iron in neurons [5]. The involvement of DMT-1 in iron homeostasis in the brain is of such significance that its alterations are linked to the onset of neurodegenerative diseases [6,7]. In the cell, iron is either used for metabolic processes, stored into Ferritin (Ftn), or exported by Ferroportin (Fpn), which has recently been described as a major contributor to iron accumulation in a cellular model of PD [8]. Given the central role of iron metabolism and oxidative stress in PD, strategies to modulate iron metabolism offer potential avenues for neuroprotection.

Recently, much attention has been given to Lactoferrin (Lf), a glycoprotein of innate immunity belonging to the transferrin family and exerting a plethora of activities, including anti-inflammatory, antioxidant and iron homeostasis-related activities [9,10,11]. The capacity of Lf in maintaining ROS balance is exerted by multiple mechanisms, including free iron chelation; the regulation of iron protein expression [12,13,14,15]; the stimulation of antioxidant proteins, such as Glutathione Peroxidase (GPx) and Superoxide Dismutase (SOD) [16,17]; and the suppression of inflammation [18]. The ability of this glycoprotein to cross the blood–brain barrier [19] makes it a promising candidate for neuroprotection strategies.

Lf naturally occurs with an iron saturation degree of about 10–20%, in a form called native Lf (Nat-Lf), but it can be easily saturated with iron to yield the so-called Holo-Lf, where more than 95% of the protein is filled with the metal. The two forms are characterized by specific structures, which confer distinct biochemical properties and therapeutic potential. Numerous studies have functionally compared the two forms, highlighting their common or distinct functions as well as their ability to modulate shared or divergent signaling pathways [20,21]. While Nat-bLf (the bovine form of Nat-Lf) is predominantly associated with antioxidant and anti-inflammatory functions [18], the role of Holo-bLf in modulating iron metabolism raised some concerns regarding its potential to exacerbate oxidative stress under certain conditions [17,22].

Recent studies have also suggested multiple roles of Lf in neuronal differentiation and neuroprotection, as it potentially modulates key processes such as neurogenesis and neuronal maturation [23,24]. Both forms of Lf promote neuronal differentiation in neuroblastoma cell lines, enhancing the expression of neuronal markers such as β-tubulin III and neurofilaments [25]. Various studies also investigated the neuroprotective role of Lf in vitro and in vivo PD models [26,27,28,29,30]. While most reports on the neuroprotective effects of Lf have focused on 1-methyl-4-phenyl-1,2,3,6-tetrahydropyridine (MPTP)-induced models of PD [26,27,28], only few studies have investigated its potential in the parkinsonian phenotype induced by rotenone [29,30]. This toxin-induced experimental model is of particular interest since, in contrast to other PD models generated by genetic manipulation or exposure to environmental pesticides, rotenone reproduces most of the PD features, including the intracellular inclusions of α-synuclein [31]. An in vivo study demonstrated that recombinant human Lf (hLf) was able to reduce rotenone-induced neurodegeneration and motor deficits in rats; however, no clues were provided on the underlying molecular mechanisms [29]. In this context, it has been recently demonstrated that hLf pre-treatment mitigates rotenone-induced toxicity by reducing oxidative stress, preserving mitochondrial integrity, and modulating apoptotic pathways in a model of neuroblastoma cells [30]. Even though these findings suggest the potential of hLf in protecting dopaminergic neurons against PD degeneration, comparative functional studies between Nat-bLf and Holo-bLf in rotenone-mediated neurodegeneration are currently absent, and the selective mechanism of action still remains largely unexplored.

Hence, the present study aims at investigating the neuroprotective efficacy of bovine Lf (bLf), in both Native and Holo forms, in a rotenone-induced in vitro PD model. While bLf shares a high degree of structural and functional homology with hLf, it differs in certain biochemical properties, including higher resistance to acidic pH and proteases, making it more suitable for oral administration than the human homolog. Specifically, we explored the differential effects of Nat-bLf and Holo-bLf on neuronal morphology, oxidative stress, iron dysregulation, and other key pathological features of PD.

## 2. Results

### 2.1. Nat-bLf Preserves Cell Viability Against Rotenone Toxicity in a Dose-Dependent Manner

First, we investigated the potential protective effect of both Native and Holo forms of bLf at different concentrations on the cell viability of differentiated N1E-115 challenged with 100 nM rotenone. The schematic experimental timeline is illustrated in Figure 1A. As expected, exposure to 100 nM of rotenone caused a pronounced reduction in cell viability compared to control (Figure 1B). Post-treatment with increasing concentrations of Nat-bLf revealed a dose-dependent protective effect, with the highest protection observed at 100 µg/mL. Conversely, Holo-bLf induced a significant recovery in cell viability only at the concentration of 100 µg/mL, whereas higher doses negatively affected cell counts. Overall, Nat-bLf markedly attenuated the cytotoxic effects of rotenone compared to Holo-bLf, highlighting a substantial difference in the biological activity of the two forms of bLf. In light of these findings, all subsequent experiments were performed using the concentration of 100 µg/mL, which exhibited the highest protective effect for both bLf forms.

### 2.2. Nat-bLf Is More Effective than Holo-bLf in Restoring Rotenone-Induced Loss of Neurites and Cytoskeletal Proteins

PD is characterized by neuronal atrophy, loss of neurites, and destabilization of microtubules [32], contributing to impaired neuronal function. To assess the impact of bLf on these alterations, we analyzed neuronal morphology and microtubule integrity in our model. No differences were recorded for Nat- and Holo-bLf treatments alone. In agreement with the Parkinsonian phenotype, rotenone exposure induced a dramatic decrease in the percentage of neurite-bearing cells (Figure 2A), with a significant neurite retraction (Figure 2B).

Post-treatment with Nat-bLf attenuated these morphological changes, increasing the number of neurite-bearing cells and significantly preserving neurite length. The morphological changes following rotenone treatment were confirmed at a molecular level by analyzing the protein expression of β3-tubulin and GAP-43, the main constituent of microtubules and the component of motile ‘growth cones’, respectively. As expected, rotenone challenge induced a significant reduction in the expression of both markers. Again, both forms of bLf were able to restore the expression of these two proteins, with Nat-bLf showing greater efficacy in preserving neuronal morphology and microtubule integrity (Figure 3).

### 2.3. bLf Preserves the Rate-Limiting Step of Dopamine Biosynthesis

Given the protective effects of bLf on neuronal morphology and microtubule integrity in rotenone-treated N1E-115 differentiated cells, we next determined whether these benefits extended to the primary pathological hallmarks of PD. The loss of dopaminergic function, a key characteristic of PD, was therefore evaluated through Western blot analysis of Tyrosine Hydroxylase (TH), a pivotal enzyme in dopamine biosynthesis. As shown in Figure 4, rotenone induced a significant reduction in TH expression. Both forms of bLf were able to normalize TH levels, with Nat-bLf exhibiting greater efficacy than Holo-bLf. These findings underline the potential of bLf to preserve dopaminergic biosynthetic machinery in neurodegenerative disorders (Figure 4).

### 2.4. bLf Counteracts SOD-1 and SOD-2 Downregulation Elicited by Rotenone

The effect of bLf in modulating antioxidant defense and counteracting oxidative stress was assessed. For this purpose, the expression of cytosolic and mitochondrial forms of the main proteins involved in the detoxification of superoxide radicals, superoxide dismutase 1 and 2 (SOD-1, SOD-2), was evaluated. Rotenone exposure significantly blunted the expression of both SOD-1 and SOD-2 (Figure 5), which was counteracted by both bLf forms, with Nat-bLf exhibiting superior efficacy. Overall, the ability of bLf to restore antioxidant defense underlines its potential to mitigate oxidative insults.

### 2.5. Nat-bLf Reduces Intracellular Oxidative Stress More Efficiently than the Holo Form

In neurodegenerative conditions, oxidative stress results from the accumulation of Reactive Oxygen species (ROS) and impairment of the endogenous antioxidant response. This imbalance contributes to cellular dysfunction through DNA damage and lipid peroxidation. To explore the potential protective role of bLf, its effects on the protection of key biological (macro)molecules were examined. The histone variant γ-H2Ax, a marker of DNA double-strand breaks, and the modified nucleotide 8-hydroxy-2′-deoxyguanosine (8-OHdG), a marker of oxidative DNA lesions, were analyzed. Both markers increased significantly upon exposure to rotenone (Figure 6A,B).

Post-treatment with Nat-bLf restored the markers to baseline levels, suggesting a robust protective effect on genomic integrity. Interestingly, Holo-bLf restored the baseline levels of γ-H2Ax (Figure 6A), but significantly increased 8-OHdG immunoreactivity, suggesting that Holo-bLf can induce an additive effect with rotenone in enhancing 8-OHdG levels (Figure 6B). Of note, 8-OHdG levels were also elevated following treatment with Holo-bLf alone, indicating a pro-oxidative impact of the saturated bLf independent of rotenone exposure. Next, the production of 4-hydroxynonenal (4-HNE), an aldehyde derived from lipid peroxidation, was analyzed. Again, rotenone exposure significantly enhanced the levels of 4-HNE (Figure 6C). Differently from Holo-bLf, post-treatment with Nat-bLf significantly attenuated this increase. Overall, these results suggest that Nat-bLf counteracts oxidative stress by reducing DNA damage and lipid peroxidation, whereas Holo-bLf was found to be ineffective.

### 2.6. Nat-bLf and Holo-bLf Differently Restore Iron Homeostasis in Rotenone-Treated Cells

Disruption of iron homeostasis is a distinctive feature of PD and contributes to oxidative stress and neurodegeneration. Prussian blue staining was used to analyze the impact of bLf on intracellular iron accumulation (Figure 7A).

In order to obtain reliable iron staining with this assay, cells were pre-treated with 100 µM Ferric Ammonium Citrate (FAC) for 24 h. The results showed that rotenone-treated cells had higher levels of intracellular iron. In contrast, cells treated with both forms of bLf showed a marked reduction of iron staining, with Nat-bLf able to almost completely eliminate intracellular iron deposition, independently from rotenone treatment (Figure 7A). These results highlight a dynamic regulation of iron within cells, where iron readily enters, as observed in the control condition, accumulates further following rotenone treatment, and is actively mobilized by both forms of bLf in the presence or absence of rotenone.

Since bLf has been widely described to be a fine orchestrator of iron homeostasis [33], the main proteins involved in brain iron mobilization were analyzed. Western blot analysis revealed that rotenone exposure led to a reduction in DMT-1 and Fpn expression, the main regulators of brain iron import and release, respectively (Figure 7B,C). Post-treatment with both forms of bLf significantly restored DMT-1 expression, with Nat-bLf exhibiting higher efficacy (Figure 7C). Interestingly, both Nat-bLf and Holo-bLf promoted Fpn overexpression, both in the absence and presence of rotenone. These findings, in agreement with the Prussian blue staining results, suggest that bLf can reduce iron accumulation by enhancing iron export, thereby mitigating iron-associated neurotoxicity.

To further investigate the differential effects of Nat- and Holo-bLfs on iron homeostasis and oxidative stress upon rotenone exposure, an additional experiment was performed to assess the levels of intracellular labile iron (Figure 8). Although the Prussian blue staining and the analyses of Fpn and DMT-1 expression suggest a potential protective role of both bLf forms in maintaining iron homeostasis, the intracellular labile iron pool remains one of the main factors capable of inducing, or exacerbating, oxidative stress, especially under conditions of mitochondrial impairment such as those induced by rotenone. The assay was based on the use of Calcein-AM, a fluorescent probe whose signal is quenched by free iron; therefore, fluorescence intensity is inversely proportional to the amount of labile iron present within the cells. The results showed that Nat-bLf–treated cells exhibited slightly lower levels of labile iron compared to control cells, whereas treatment with Holo-bLf alone led to a marked increase in intracellular free iron concentration relative to the control. This increase may account, at least in part, for the pro-oxidant effects observed through 8-OHdG visualization (Figure 6B) and, consequently, for the lower efficacy of the Holo form in restoring cell viability (Figure 1B). Rotenone-treated cells displayed elevated levels of labile iron compared to the control, which remained high upon co-treatment with Holo-bLf. In contrast, co-treatment with the native form slightly, although not significantly, reduced intracellular iron levels, likely due to its preserved iron-chelating capacity, which is lost in the Holo form.

### 2.7. Nat-bLf Prevents α-Synuclein Intracellular Accumulation

Finally, the accumulation of α-synuclein was assessed by immunofluorescence analysis. Figure 9 shows that rotenone treatment resulted in a significant buildup of α-synuclein when compared to control cells. Nat-bLf treatment significantly reduced α-synuclein immunoreactivity, highlighting its neuroprotective role in hindering protein accumulation and aggregation. In contrast, Holo-bLf exacerbated α-synuclein levels, even in the absence of rotenone. This suggests that the Holo-form of bLf may promote aggregation and/or defective clearance under specific conditions.

## 3. Discussion

Although several papers have already investigated the beneficial effects of Lactoferrin in different PD models [26,27,28,29,30], the present study is the first to highlight the neuroprotective potential of different forms of bLf, namely the Native and the Holo forms, against rotenone-induced toxicity in an in vitro model of PD. The ability of rotenone to induce systemic neurotoxicity, progressive dopaminergic degeneration, mitochondrial dysfunction, and protein aggregation, alongside its epidemiological relevance, makes it the most comprehensive and reliable model for studying the pathogenesis of PD in both in vitro and in vivo systems [31,34,35].

First, our study explored a relatively under-researched area: the role of Lf in maintaining neuronal morphology. Neuronal atrophy and microtubule destabilization are well-documented consequences of PD pathology [32]. Few studies have focused on the preservation of microtubule integrity in PD, yet this research reveals how Nat-bLf could offer a novel strategy for stabilizing the cytoskeleton. The ability of Nat-bLf to significantly restore neurite network aligns with findings that Lf supports neuronal differentiation [25] and cytoskeletal remodeling [36]. The molecular assessment via β3-tubulin and GAP-43 expression confirms that Nat-bLf effectively preserves microtubule integrity, whereas Holo-bLf, though beneficial, is less effective. Further, the restoration of TH expression by both bLf forms, with Nat-bLf showing superior efficacy, reinforces previous findings on the role of Lf in supporting dopaminergic neurons, through both the regulation of neurotrophic factors and the reduction in oxidative stress [27,28,37].

Oxidative stress is a major contributor to PD pathogenesis, with enzymes of the SOD family playing a crucial role in ROS detoxification [38]. The restoration of SOD-1 and SOD-2 levels upon treatment of rotenone-treated cells with both forms of bLf highlights, once again, their capability to act as positive regulators of antioxidant genes. However, when considering both oxidative DNA and lipid damage markers, Nat- and Holo-bLf exhibit distinct protective profiles against rotenone-induced toxicity, highlighting form-specific differences in their neuroprotective efficacy. Nat-bLf’s capacity to restore these markers to baseline levels underscores its role in genomic protection, possibly via direct scavenging of free radicals and enhancement of DNA repair pathways. The pro-oxidative effect of Holo-bLf, evidenced by elevated 8-OHdG levels, suggests that an iron-dependent mechanism, consistent with previous reports on ferritin- and iron-induced genotoxicity [39,40], may not be the sole factor involved in the exacerbation of oxidative DNA lesions. As a matter of fact, bLf is a multifunctional glycoprotein able to trigger a wide range of pathways and mechanisms, and its degree of iron saturation is increasingly recognized as a key determinant in modulating the downstream biological effects mediated by the glycoprotein, which remain insufficiently characterized and warrant further investigation. The 4-HNE reduction by Nat-bLf further emphasizes its protective role against lipid peroxidation, consistent with literature advocating its use in mitigating oxidative insults in neurodegenerative diseases [17]. The higher efficacy of Nat-bLf vs. Holo-bLf is particularly relevant, as previous studies suggest that iron-binding proteins in their native form exhibit enhanced ROS-scavenging capabilities compared to their Holo counterparts, also in the context of neurodegenerative processes [17,41].

In this frame, iron dysregulation is a well-documented feature of PD, where it contributes to oxidative stress and neurotoxicity [42]. Prussian blue staining demonstrates that both bLf forms can reduce intracellular iron accumulation, with Nat-bLf demonstrating greater efficacy. The restoration of DMT-1 and Fpn levels by both bLf forms further supports their role in maintaining iron homeostasis, consistent with the evidence linking Lf to its anti-inflammatory activity (IL-6 decrease which in turn increases DMT-1 and Fpn expression) and to the regulation of iron metabolism in neurodegeneration [33]. The greater efficacy of Nat-bLf in mobilizing iron may be attributed to its enhanced capacity to both chelate free iron and regulate iron transport proteins compared to the Holo form [33]. Our data highlight distinct roles of the two bLf forms in regulating cellular iron homeostasis. Both Nat-bLf and Holo-bLf reduce total intracellular iron accumulation and restore the expression of iron transport proteins. However, only the Native form effectively maintains a balanced labile iron pool, likely due to its preserved iron-binding capacity. This property may underlie the superior antioxidant efficacy of Nat-bLf observed in the rotenone model, reinforcing the idea that effective control of iron availability is critical for limiting oxidative stress and preserving neuronal integrity.

In addition, iron dysregulation is closely linked to α-synuclein fibrillation, as excessive iron facilitates misfolding and aggregation, exacerbating cellular toxicity [43]. The reduction in α-synuclein accumulation with Nat-bLf treatment is in line with studies indicating that Lf can inhibit protein aggregation, including α-synuclein in PD mice challenged with MPTP [27]. Compared to other iron chelators, such as deferoxamine, Lf uniquely combines iron homeostasis restoration with additional neurotrophic support, positioning it as a superior candidate for targeting α-synuclein pathology in PD. On the other hand, the exacerbation of α-synuclein aggregation by Holo-bLf, even in the absence of rotenone, corroborates the hypothesis of a link between oxidative stress, iron accumulation and protein aggregation [44]. Indeed, our data suggest that the pro-oxidant phenotype induced by Holo-bLf does not result from impaired iron export but rather from redox cycling of iron bound to the protein or secondary activation of oxidative signaling pathways.

The evidence collected in this study, by comparing the effects induced by Nat-bLf and Holo-bLf, highlights the importance of iron-binding capacity in determining the effectiveness of iron-related therapies. While Holo-bLf demonstrated pro-oxidative effects, Nat-bLf maintained a balance in iron homeostasis, highlighting its higher therapeutic potential for PD. In particular, under rotenone challenge, Nat-bLf displayed a more pronounced protective response, which can be ascribed both to its intrinsic ability to chelate labile iron and to its stronger positive effect on antioxidant enzymes. Together, these findings suggest that Nat-bLf promotes a coordinated antioxidant defense under oxidative stress, whereas Holo-bLf, due to its iron load, may shift the redox balance toward pro-oxidant conditions even in basal settings. With these promising findings, Nat-bLf stands out as a viable candidate for further preclinical and clinical investigations, especially when compared to traditional iron chelators and mitochondrial-targeted therapies that have shown limited success [45,46].

Overall, the present study offers novel insights into the neuroprotective effects of Nat-bLf, marking a significant step forward in understanding its potential in treating PD. Unlike conventional therapies that typically focus on specific mechanisms like antioxidant activity or iron chelation, Nat-bLf provides a comprehensive, multi-faceted approach to neuroprotection. Its effects span mitochondrial support, stabilization of the cytoskeleton, reduction in oxidative stress, regulation of iron homeostasis, and clearance of α-synuclein, which collectively contribute to its broad-spectrum protective properties. This holistic mechanism enhances the translational potential of Nat-bLf as a therapeutic candidate for PD, positioning it as an advanced adjuvant treatment to be combined with standard therapies.

Recently, a study by Yong and colleagues has investigated the role of hLf in a rotenone-induced PD in vitro model using SH-SY5Y cells. Although the iron-saturation rate of hLf was not taken into account, and the experimental plan was based on the hLf pre-treatment of cells before rotenone insult, both the antioxidant and anti-apoptotic effects of hLf were demonstrated [30].

To date, bLf is considered more reliable than hLf for therapeutic applications in neurodegenerative diseases due to its higher stability, bioavailability, and iron-binding properties. Unlike hLf, bLf is more resistant to acidic pH and enzymatic degradation, resulting in better gastrointestinal stability and improved oral bioavailability [47], which is crucial for non-invasive treatment strategies and presents a stronger iron-binding affinity, making it more effective in regulating iron homeostasis in neurodegenerative disorders [33]. Moreover, bLf is more cost-effective and scalable in production since it is abundantly available in cow’s milk, whereas hLf requires recombinant technology for large-scale manufacturing, usually not ensuring the production of a glycoprotein embedded with the same physico-chemical properties of the native one, due to differences in iron-saturation rate, glycosylation patterns, purity and stability [9]. Importantly, bLf has been extensively used in human nutrition, including infant formulas and dietary supplements [48], with well-documented safety profiles, further supporting its feasibility for long-term therapeutic use in neurodegenerative diseases. Importantly, the therapeutic relevance of bLf extends beyond its antioxidant and iron-regulating properties. Its ability to cross the blood–brain barrier [19], a critical requirement for effective neurodegenerative therapies, highlights its promise as a neuroprotective agent capable of acting directly within the central nervous system. These characteristics collectively position bLf as a more promising candidate than hLf for neuroprotective interventions.

However, despite these encouraging results, the study has certain limitations that need to be addressed in future research. The in vitro models used, while valuable for uncovering mechanistic insights, lack the complexity and physiological relevance of in vivo systems. Additionally, while the study provides evidence of short-term neuroprotection, long-term studies are essential to determine whether continuous administration of Nat-bLf can prevent progressive neurodegeneration over time.

In summary, this study lays the foundation for the exploration of Nat-bLf as a promising therapeutic agent capable of counteracting disorders associated with oxidative stress and mitochondrial dysfunction, including neurodegenerative diseases, while also highlighting key areas that require further investigation to fully understand its therapeutic potential.

## 4. Materials and Methods

### 4.1. Bovine Lactoferrin

Bovine Lactoferrin (bLf) was obtained as a highly purified powder (purity > 99%, determined by SDS-PAGE and Coomassie Blue staining) from Vivatis Pharma, Gallarate, Italy. The concentration of bLf solutions was measured via UV spectroscopy using an extinction coefficient of 15.1 for a 1% solution at 280 nm. Protein purity was about 99%, as checked by SDS-PAGE and silver nitrate staining. Iron saturation of the native bLf (Nat-bLf) was approximately 15%, as confirmed by absorbance at 468 nm with an extinction coefficient of 0.54 for a 1% solution of fully iron-saturated protein. Endotoxin contamination was assessed using the Limulus Amebocyte Lysate (LAL) assay (Pyrochrome kit, PBI International, Milan, Italy) and found to be below 0.5 ng/mg. Before experimental use, bLf solutions were sterilized through a 0.2 µm low protein retention filter (Millex HV, Millipore, Burlington, MA, USA).

### 4.2. Preparation of Holo-bLf

Holo-bLf was prepared by saturating Nat-bLf with ferric ions. Briefly, Nat-bLf was dissolved at 20 mg/mL in 0.1 M sodium bicarbonate buffer and incubated with 5 mM ferric citrate under gentle stirring for 2 h at room temperature. Excess unbound iron was removed by dialysis against 0.1 M sodium bicarbonate for 48 h, with regular buffer changes. The resultant Holo-bLf solution, controlled by absorbance at 468 nm, showed an iron saturation > 95%, and it was aliquoted, frozen, and stored at −20 °C.

### 4.3. Cell Culture and Differentiation

Mouse neuroblastoma clone N1E-115 was obtained from the European Collection of Cell Cultures (Cat. no. 88112303) (Salisbury, UK) at passage number 20 at the start of the study. N1E-115 neuroblastoma cells were cultured in Dulbecco’s Modified Eagle Medium (DMEM) supplemented with 10% Fetal Bovine Serum (FBS), 100 U/mL penicillin, and 100 µg/mL streptomycin, under standard conditions (37 °C, 5% CO_2_). To induce neuronal differentiation, cells were cultured in DMEM containing 0.5% FBS for 96 h, which was confirmed by morphological changes indicative of neurite outgrowth.

We selected the N1E-115 neuroblastoma cell line, which exhibits a partially dopaminergic phenotype due to its ability to synthesize DOPA and, to a lesser extent, dopamine and noradrenaline. Notably, this cell line has the capacity to differentiate into neuron-like cells exhibiting dopaminergic characteristics relevant to Parkinson’s disease [49,50,51]. Using a single, consistent model enabled precise analysis of bLf effects on key PD-related pathways, including oxidative stress, iron homeostasis, and α-synuclein accumulation. While we acknowledge the limitations of single-model studies, this approach provides a focused foundation for future validation in additional cellular and in vivo systems.

### 4.4. Induction of Parkinson’s Disease Phenotype and Treatments

N1E-115 neuroblastoma cells were seeded in 6-multiwell plates at a density of 1.5 × 10^5^ per well in 2 mL of DMEM containing 10% FBS, 100 U/mL penicillin, and 100 µg/mL streptomycin, under standard conditions (37 °C, 5% CO_2_). After 6 h, in order to induce differentiation, medium was changed with fresh DMEM containing 0.5% FBS, 100 U/mL penicillin, and 100 µg/mL streptomycin (37 °C, 5% CO_2_). After 72 h of incubation, medium was refreshed (0.5% FBS) and, at 96 h, differentiated N1E-115 cells were treated with 100 nM rotenone, since this dosage falls into the range commonly used to induce the parkinsonian phenotype in different cell culture models [52,53,54]. Following 3 h of rotenone exposure, and without removing the medium, cells were treated for an additional 24 h with different concentrations of either Nat-bLf or Holo-bLf (10, 50, 100, 250, 500 and 1000 µg/mL) for cell counts, and with the sole dose of 100 µg/mL for both bLfs in all subsequent experiments. The 100 µg/mL dose of bLf was chosen following the results obtained from the cell counts experiments. Of note, such a dose is usually employed in antioxidant studies without cytotoxic effects [17,55], and this concentration of hLf is also observed in the bloodstream during infection and inflammation, thereby reflecting physiologically relevant conditions [56]. Untreated cells were used as controls. The experimental scheme is presented in Figure 1A.

### 4.5. Cell Counts and Morphological Analysis

The analysis of neuronal morphology in N1E-115 cells was assessed by acquiring images with an inverted microscope. Cell counts and quantitative analyses were performed on images acquired at 10× magnification, which allows accurate assessment of overall culture density and neurite network organization. Morphometric analysis was performed using ImageJ software (version 1.52a), quantifying the total number of cells (expressed as a percentage change from control), the mean neurite length (expressed as a percentage of control), and the rate of neurite-bearing cells. The percentage of neurite-bearing cells was determined by calculating the ratio of differentiated cells to the total number of cells per field. N1E-115 cells were considered differentiated if they had at least one neurite of length equal to or greater than the diameter of the soma. For each experimental condition, at least three independent experiments were performed, each comprising triplicate wells, with a minimum of five randomly acquired fields per well analyzed.

### 4.6. Immunofluorescence Analysis

Cells (1.2 × 10^5^ per well) were seeded onto glass coverslips pre-coated with poly-L-lysine to promote cell adhesion. Coverslips were placed in 6-well plates and cells were treated as described in Section 4.4. At the end of the experiment, cells were fixed with 4% paraformaldehyde (PFA) in PBS for 15 min and permeabilized with 0.1% Triton X-100 in PBS. Cells were incubated with 3% Bovine Serum Albumin (BSA) to block nonspecific binding for 1 h at room temperature. Primary antibodies against 8-hydroxy-2′-deoxyguanosine (8-OHdG) (sc66036, Santa Cruz, CA, USA) (1:100), 4-hydroxynonenal (4-HNE) (Thermo Fisher, Waltham, MA, USA, MA527570) (1:50), and α-synuclein (sc12767 Santa Cruz, CA, USA) (1:50 dilution) were applied overnight at 4 °C. Afterward, cells were incubated with fluorophore-conjugated secondary antibodies (Alexa Fluor™ 546 and Alexa Fluor™ 488, Invitrogen, Waltham, MA, USA) for 1 h at room temperature. Nuclei were stained with DAPI for visualization. Images were acquired using the Leica TCS SP8 confocal microscope and Leica Application Suite X (LAS X) software (version 3.5.5) at 40× magnification.

### 4.7. Perl’s Prussian Blue Staining

Cells seeded in six wells were treated with 100 µM Ferric Ammonium Citrate (FAC) after 72 h of differentiation. Following 48 h, the cells were washed three times with Phosphate-Buffered Saline (PBS) and stained with Perl’s Prussian blue stain, consisting of potassium ferrocyanide and hydrochloric acid (HCl). The solution is prepared by mixing a 2% K_4_Fe(CN)_6_ solution with a 2% HCl solution in a 1:1 ratio. It should be freshly prepared before use by diluting the 20% stock solutions of K_4_Fe(CN)_6_ and HCl. The reaction produces blue granules upon interacting with ferritin in the cells, enabling the identification and localization of iron deposits. Images were acquired using inverted microscope 20× objective.

### 4.8. Quantification of Cellular Labile Iron Pool

Relative changes in intracellular labile iron were assessed using the Calcein–acetoxymethyl ester (Calcein-AM, 17783, Sigma-Aldrich, Milan, Italy) fluorescence method. This assay mainly reflects the cytosolic labile iron pool; moreover, changes in Calcein fluorescence also provide an indirect indication of oxidative stress, since alterations in the labile iron pool are closely linked to ROS generation and redox imbalance, as previously reported [57].

N1E-115 cells were seeded at a density of 2 × 10^5^ cells on glass coverslips and cultured as described in Figure 1A. After 120 h of treatment with rotenone and/or Nat- or Holo-bLf, cells were incubated with 10 µM Calcein-AM for 20 min at 37 °C in the dark. Following two washes with PBS, cells were maintained in PBS for an additional 15 min at 37 °C to allow complete de-esterification of the probe. Images were acquired using the Leica TCS SP8 confocal microscope and Leica Application Suite X (LAS X) software at 40× magnification.

### 4.9. Western Blotting

Protein expression levels were assessed using Western blotting. Total protein lysates were prepared in a buffer containing 25 mM MOPS (pH 7.4), 150 mM NaCl, and 1% Triton X-100, supplemented with protease inhibitors (1 mM PMSF, 2 μM leupeptin, and 2 μM pepstatin) and incubated on ice for 1 h. Protein concentrations were determined using the Bradford assay. Equal amounts of protein (20–30 µg) were resolved on SDS-polyacrylamide gels and transferred onto nitrocellulose membranes. The membranes were blocked with 5% milk prepared in PBS containing 0.1% Tween-20 (PBST) for 1 h. Subsequently, they were incubated overnight at 4 °C with primary antibodies. For Western blot analysis, the following primary antibodies were employed: anti-β-3-Tubulin (sc-80005, Santa Cruz, CA, USA) (1:1000), anti-Growth-Associated Protein-43(GAP-43) (sc17790, Santa Cruz, CA, USA) (1:1000), anti-Fpn 31A5, (Amgen, Thousand Oaks, CA, USA) (1:1000), anti-DMT-1 (sc-166884, Santa Cruz, CA, USA) (1:1000), anti-SOD-1 (sc-17767, Santa Cruz, CA, USA), anti-SOD-2 (sc-137254, Santa Cruz, CA, USA), anti- *p*-Histone H2A.X (Ser 139) (sc-517348, Santa Cruz, CA, USA) (1:1000), anti-GAPDH (sc-47724, Santa Cruz, CA, USA) (1:1000), anti-β-Tubulin (sc-101527, Santa Cruz, CA, USA) (1:5000) and anti-β-Actin (sc1616, Santa Cruz, CA, USA) (1:1000). After incubation with the appropriate secondary horseradish peroxidase-conjugated antibody, blots were developed with Clarity Western ECL substrate (170-5061, Biorad, Milan, Italy). Signals were detected using an enhanced chemiluminescence system (Bio-Rad ChemiDoc, Milan, Italy) and quantified using ImageJ software (version 1.52a).

### 4.10. Statistical Analysis

Statistical analyses were conducted using GraphPad Prism software 9.0. Statistical significance was determined using one-way ANOVA followed by Tukey’s multiple comparison test to assess differences among multiple independent groups, appropriate for the experimental design. Data are presented as mean ± SEM.

## 5. Conclusions

This study provides compelling evidence that the neuroprotective efficacy of bLf in Parkinson’s disease models critically depends on its iron-saturation state. Nat-bLf demonstrated robust protection against rotenone-induced neurotoxicity by preserving neuronal morphology, stabilizing microtubules, restoring dopaminergic markers, and enhancing antioxidant defenses, particularly through the upregulation of mitochondrial SOD-2. In contrast, Holo-bLf, burdened by its iron load, augmented iron labile pool, elicited pro-oxidative responses and aggravated α-synuclein aggregation, underscoring the detrimental impact of excessive iron saturation on the antioxidant activity of the glycoprotein.

The superior performance of Nat-bLf in mitigating oxidative stress, restoring iron homeostasis, and preventing protein aggregation positions it as a multifunctional therapeutic agent capable of addressing key pathological processes in PD. Its inherent stability, bioavailability, and safety profile further strengthen its translational value compared to other iron chelators or antioxidant therapies.

Together, these findings identify Nat-bLf as a promising, physiologically compatible candidate for future neuroprotective strategies, warranting in vivo validation and clinical exploration to harness its full potential in combating PD and related neurodegenerative disorders.

## Figures and Tables

**Figure 1 ijms-26-11312-f001:**
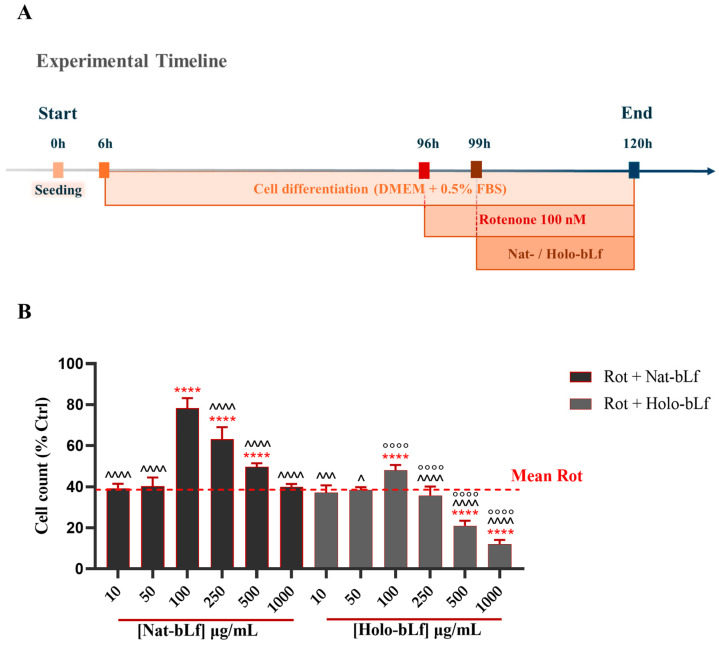
Quantitative analysis of cell viability. (**A**) Schematic illustration of the experimental timeline. (**B**) Cell viability of N1E-115 differentiated cells treated with 100 nM rotenone (Rot), in the presence or absence of different concentrations of Native (Nat)-bLf or Holo-bLf, for 24 h. Statistical significance was determined using one-way ANOVA followed by Tukey’s multiple comparison test and data were expressed as mean ± standard error of the mean (SEM). For each experimental condition, at least three independent experiments were performed, each comprising triplicate wells, with a minimum of five randomly acquired fields per well analyzed. **** v Rot 100 nM (*p* < 0.0001); ^^^^ v bLf 100 µg/mL (*p* < 0.0001); ^^^ v bLf 100 µg/mL (*p* < 0.001); ^ v bLf 100 µg/mL (*p* < 0.05); °°°° v the same concentration of Nat-bLf (*p* < 0.0001).

**Figure 2 ijms-26-11312-f002:**
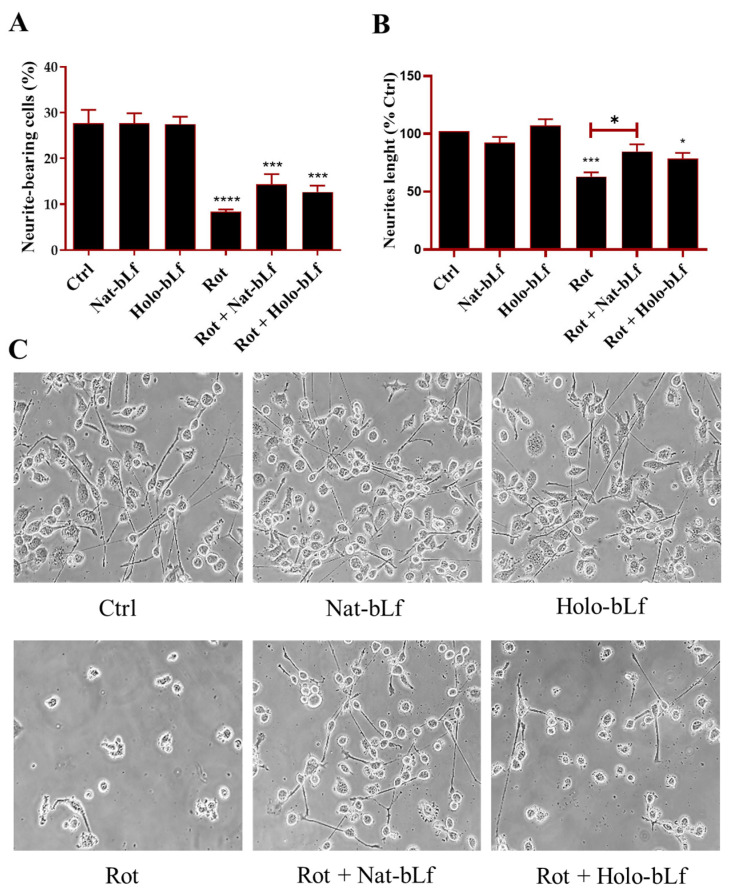
Assessment of neuronal morphology. Bright field images and neuronal morphology assessment of N1E-115 differentiated cells treated with 100 nM rotenone (Rot), in the presence or absence of 100 µg/mL of Nat-bLf or Holo-bLf, for 24 h. The percentage of neurite-bearing cells (**A**) and neurite length (**B**) were quantified using ImageJ software (version 1.52a). Analyses were performed on images acquired at 10× magnification, while representative 40× images are here shown to illustrate neurite morphology and branching (**C**). Statistical significance was determined using one-way ANOVA followed by Tukey’s multiple comparison test and data were expressed as mean ± standard error of the mean (SEM). For each experimental condition, at least three independent experiments were performed, each comprising triplicate wells, with a minimum of five randomly acquired fields per well analyzed. Asterisks (*) indicate a statistically significant difference compared to the control group; horizontal bars with asterisks indicate a significant difference between specific groups (**** *p* < 0.0001; ***: *p* < 0.001; *: *p* < 0.05).

**Figure 3 ijms-26-11312-f003:**
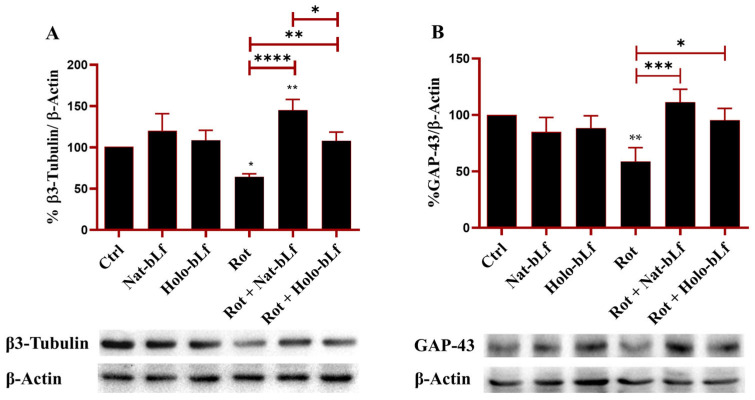
Protein expression analysis of β3-tubulin and Growth Associated Protein (GAP)-43. Western blot and densitometric analysis of β3-tubulin (**A**) and GAP-43 (**B**) in N1E-115 differentiated cells treated with 100 nM rotenone (Rot), in presence or absence of 100 µg/mL of Nat-bLf or Holo-bLf, for 24 h. Protein expression levels were normalized to β-Actin. Statistical significance was determined using one-way ANOVA followed by Tukey’s multiple comparison test and data were expressed as mean ± standard error of the mean (SEM). *n* = 3 different experiments. Asterisks (*) indicate a statistically significant difference compared to the control group; horizontal bars with asterisks indicate a significant difference between specific groups (**** *p* < 0.0001; ***: *p* < 0.001; **: *p* < 0.01; *: *p* < 0.05).

**Figure 4 ijms-26-11312-f004:**
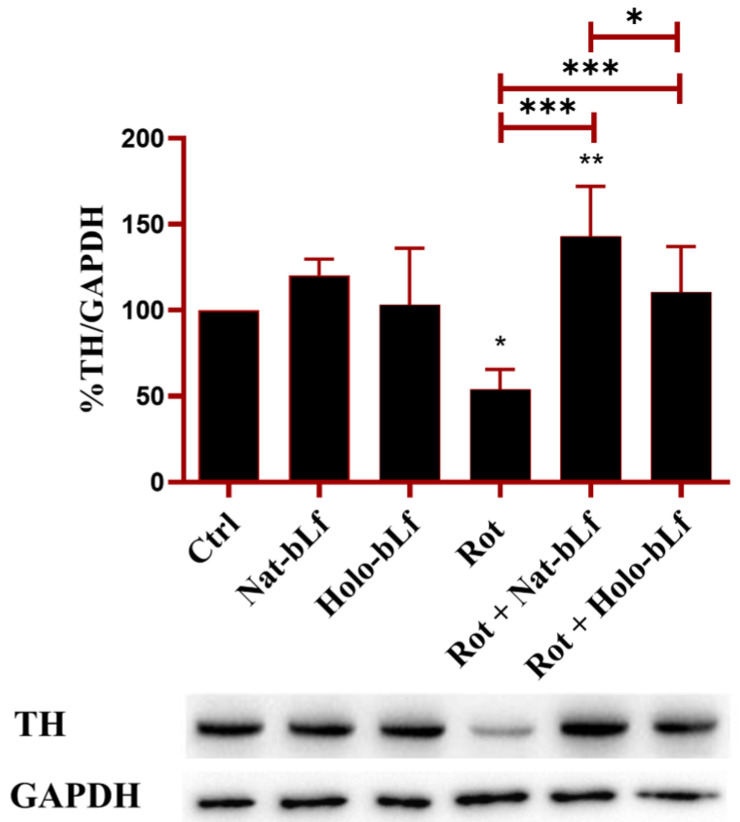
Protein expression analysis of Tyrosine Hydroxylase (TH). Western blot and densitometric analysis of TH protein levels in N1E-115 differentiated cells treated with 100 nM rotenone (Rot), in presence or absence of 100 µg/mL of Nat-bLf or Holo-bLf, for 24 h. Protein expression levels were normalized to GAPDH. Statistical significance was determined using one-way ANOVA followed by Tukey’s multiple comparison test and data were expressed as mean ± standard error of the mean (SEM). *n* = 3 different experiments. Asterisks (*) indicate a statistically significant difference compared to the control group; horizontal bars with asterisks indicate a significant difference between specific groups (***: *p* < 0.001; **: *p* < 0.01; *: *p* < 0.05).

**Figure 5 ijms-26-11312-f005:**
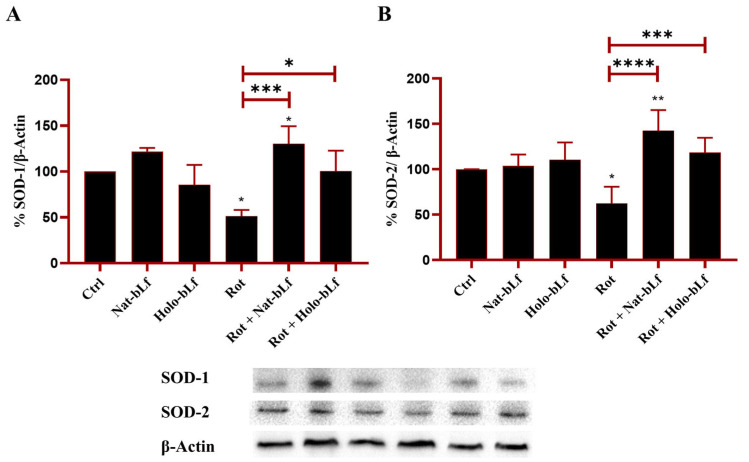
Protein expression analysis of Superoxide Dismutase (SOD)-1 and SOD-2. Western blot and densitometric analysis of SOD-1 (**A**) and SOD-2 (**B**), in N1E-115 differentiated cells treated with 100 nM rotenone (Rot), in presence or absence of 100 µg/mL of Nat-bLf or Holo-bLf, for 24 h. Protein expression levels were normalized to β-Actin. Statistical significance was determined using one-way ANOVA followed by Tukey’s multiple comparison test and data were expressed as mean ± standard error of the mean (SEM). *n* = 3 different experiments. Asterisks (*) indicate a statistically significant difference compared to the control group; horizontal bars with asterisks indicate a significant difference between specific groups (**** *p* < 0.0001; ***: *p* < 0.001; **: *p* < 0.01; *: *p* < 0.05).

**Figure 6 ijms-26-11312-f006:**
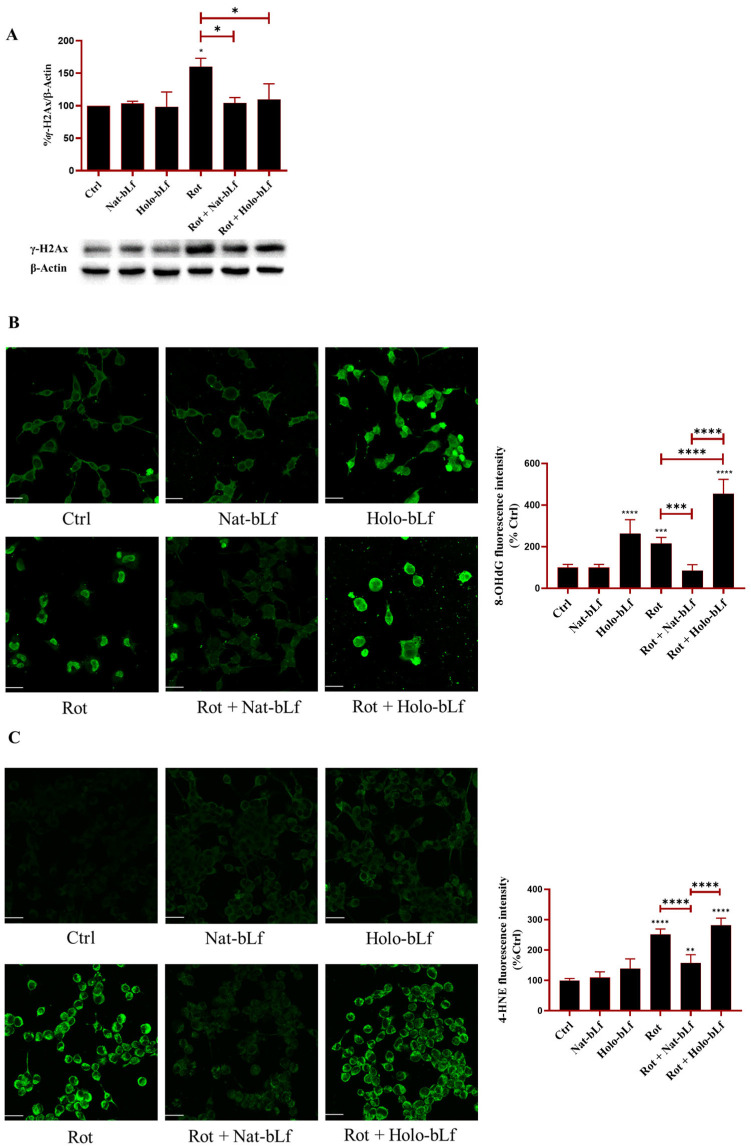
Analysis of oxidative stress markers. Western blot analysis of DNA damage marker γ-H2AX (**A**) and confocal analysis of 8-hydroxy-2′-deoxyguanosine (8-OHdG) (**B**) and 4-hydroxynonenal (4-HNE) (**C**) expression in N1E-115 differentiated cells treated with 100 nM rotenone (Rot), in the presence or absence of 100 µg/mL Nat-bLf or Holo-bLf for 24 h. Protein expression levels were normalized to β-Actin. Immunofluorescence images were acquired using the Leica TCS SP8 confocal microscope and Leica Application Suite X (LAS X) software (version 3.5.5) at 40× magnification. Scale bar: 10 μm. Quantification of fluorescence intensity was performed using ImageJ analysis software (version 1.52a). Statistical significance was determined using one-way ANOVA followed by Tukey’s multiple comparison test and data were expressed as mean ± standard error of the mean (SEM). *n* = 3 different experiments for Western blot and *n* = 5 for immunofluorescence analysis. Asterisks (*) indicate a statistically significant difference compared to the control group; horizontal bars with asterisks indicate a significant difference between specific groups (**** *p* < 0.0001; ***: *p* < 0.001; **: *p* < 0.01; *: *p* < 0.05).

**Figure 7 ijms-26-11312-f007:**
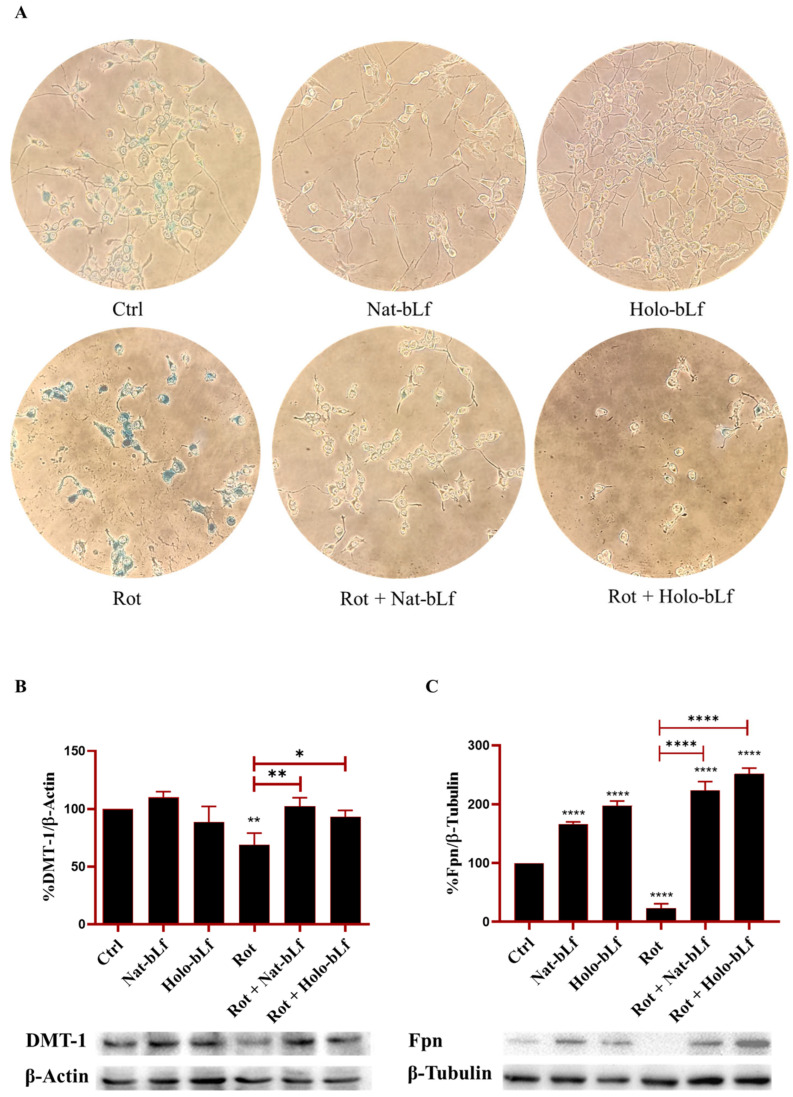
Analysis of intracellular iron content and expression of iron transporter proteins. Perl’s Prussian Blue staining (**A**), Western blot and densitometric analysis of Divalent Metal Transporter-1 (DMT-1) (**B**) and Ferroportin (Fpn) (**C**) protein expression in N1E-115 differentiated cells treated with 100 nM rotenone (Rot), in the presence or absence of 100 µg/mL of Nat-bLf or Holo-bLf, for 24 h. Protein expression levels were normalized to β-Actin. Statistical significance was determined using one-way ANOVA followed by Tukey’s multiple comparison test and data were expressed as mean ± standard error of the mean (SEM). *n* = 4 different experiments for Perl’s Prussian blue and *n* = 3 for Western blot. Asterisks (*) indicate a statistically significant difference compared to the control group; horizontal bars with asterisks indicate a significant difference between specific groups (**** *p* < 0.0001; **: *p* < 0.01; *: *p* < 0.05).

**Figure 8 ijms-26-11312-f008:**
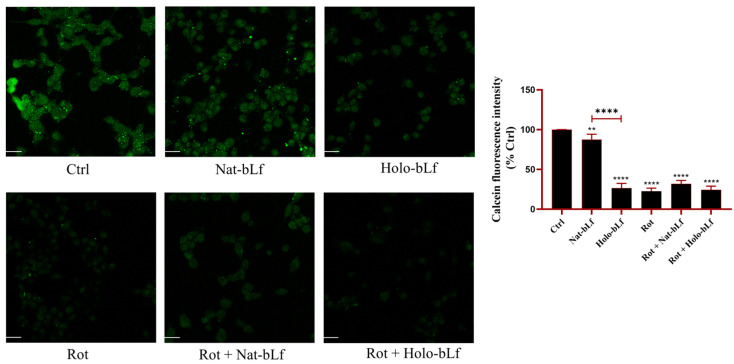
Confocal analysis of Calcein-AM. Confocal analysis of Calcein-AM in N1E-115 differentiated cells treated with 100 nM rotenone (Rot), in presence or absence of 100 µg/mL of Nat-bLf or Holo-bLf, for 24 h. Images were acquired using the Leica TCS SP8 confocal microscope and Leica Application Suite X (LAS X) software (version 3.5.5) at 40× magnification. Scale bar: 10 μm. Quantification of fluorescence intensity was performed using ImageJ analysis software (version 1.52a). Statistical significance was determined using one-way ANOVA followed by Tukey’s multiple comparison test and data were expressed as mean ± standard error of the mean (SEM). *n* = 3 different experiments. Asterisks (*) indicate a statistically significant difference compared to the control group; horizontal bars with asterisks indicate a significant difference between specific groups (**** *p* < 0.0001; **: *p* < 0.01).

**Figure 9 ijms-26-11312-f009:**
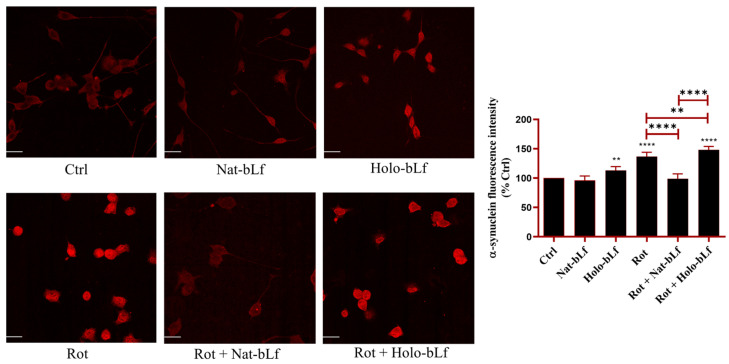
Confocal analysis of α-synuclein expression. Confocal analysis of α-synuclein expression in N1E-115 differentiated cells treated with 100 nM rotenone (Rot), in presence or absence of 100 µg/mL of Nat-bLf or Holo-bLf, for 24 h. Images were acquired using the Leica TCS SP8 confocal microscope and Leica Application Suite X (LAS X) software (version 3.5.5) at 40× magnification. Scale bar: 10 μm. Quantification of fluorescence intensity was performed using ImageJ analysis software (version 1.52a). Statistical significance was determined using one-way ANOVA followed by Tukey’s multiple comparison test and data were expressed as mean ± standard error of the mean (SEM). *n* = 6 different experiments. Asterisks (*) indicate a statistically significant difference compared to the control group; horizontal bars with asterisks indicate a significant difference between specific groups (**** *p* < 0.0001; **: *p* < 0.01).

## Data Availability

The data presented in this study are available in the current article.

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
