# Peer review of "Exploring the Multifaceted Neuroprotective Mechanisms of Bovine Lactoferrin in a Cell Culture Model of Parkinson’s Disease"

_ijms, 2025, doi:10.3390/ijms262311312_

Round 1

Reviewer 1 Report

Comments and Suggestions for Authors

Dear authors, please see the attached file.

Author Response

We thank the Reviewer for their helpful comments. The paper has been amended accordingly and a point-to-point reply follows.

An article written by Giusi Ianiro and colleagues, entitled Exploring the multifaceted neuroprotective mechanisms of bovine Lactoferrin in a cell culture model of Parkinson’s Disease. Indeed, the fight against neurodegenerative diseases is one of the priority challenges for modern society. The aim of this study was to evaluate the effectiveness of using two forms of bovine lactoferrin (native and iron-rich forms) as a neuroprotective agent in Parkinson’s disease (PD). The studies were conducted in vitro, on the N1E-115 cell line differentiated into a neuronal phenotype. Rotenone was used as a toxin to model PD. It was shown that in some cases Nat-Lf and Holo-Lf contributed to the restoration of cells after exposure to rotenone at a concentration of 100 nM.

However, after carefully studying the article and the data presented in it, I had a number of questions regarding the idea, methods and approach to completing the tasks.

1. Using the N1E-115 line. The authors provide references where this line was used, but none of these works directly state that this line can be used as a PD model. Researchers who used this line studied various processes associated with either enzymes, or the cytoskeleton, or the effect of growth factors. Considering that N1E-115 is an adrenergic cell clone of mouse neurons, it is more correct to indicate that these are cells that partially possess a dopaminergic phenotype.

Reply: We thank the reviewer for this observation, which gave us the opportunity to better justify the use of our experimental model. As reported by the reviewer, N1E-115 cells are largely employed to study differentiation, enzyme activity as well as cytoskeleton remodeling (DOI: 10.1007/s00726-005-0256-z; DOI: 10.1111/j.1471-4159.1973.tb07568.x). Despite this, N1E-115 cells are also used as a valuable cell model to study the molecular mechanisms and the putative therapeutic targets in Parkinson’s disease (DOI: 10.3389/fnmol.2017.00447; DOI: 10.1016/j.neuroscience.2009.10.042; DOI: 10.1371/journal.pone.0008268; DOI: 10.3390/ijms23063110). Notably, as stated by the reviewer, N1E-115 cells partially possess a dopaminergic phenotype, given their ability to synthetize DOPA and, to a lesser extent, dopamine and noradrenaline (DOI: 10.1016/0006-2952(82)90016-8). In order to be more accurate in the description of the selected cell model, we revised the manuscript to highlight the partial dopaminergic nature of N1E-115 cells as well as their use in research on Parkinson’s disease.

2. The description of differentiation and experiments is not complete and does not allow for a clear interpretation of how the researchers conducted their work. For example, Line 428 – 429: After 3 hours of rotenone exposure, cells were treated with 100 μg/mL of either Nat-bLf or Holo-bLf for an additional 24 hours. It is not clear whether after 3 hours of incubation with Rotenone, Nat-bLf or Holo-bLf was simply added to the medium. Or was the medium changed and incubation with Nat-bLf or Holo-bLf was carried out without the presence of rotenone? I recommend that the authors draw a diagram of the experiments. This will allow the reader to understand how and in what sequence the stages of the experiments were carried out.

Reply: We thank the reviewer for pointing out the absence of these details in the text, allowing us to better specify the experimental design. After 3 hours of rotenone exposure, without removing the medium, Nat‑bLf or Holo‑bLf were directly added to the rotenone‑containing medium, thus maintaining co‑treatments for an additional 24h. We have clarified this point in the Materials and Methods 4.4 section. Further, a diagram with the experimental timeline has been added to the main text as Figure 1A.

3. There is no explanation why the concentration of 100 nM was taken. In addition, the authors are convinced that the quote "low dose rotenone (100 nM) to induce a progressive Parkinsonian phenotype and mitochondrial dysfunction without causing acute toxicity or uncontrolled cell death". However, according to the results presented in the article, 3-hour treatment with 100 nM rotenone leads to the death of 60% of cells (+-10%). This is acute death of cells. In other articles, the concentrations and effects on cells vary greatly from each other, so this explanation is very important.

Reply: We appreciate the reviewer’s comment, which helped us to better justify our rationale about the dose of rotenone. For our experiments, we selected 100 nM of rotenone since this dosage falls into the range commonly used to induce the parkinsonian phenotype in different cell culture models (DOI: 10.1016/j.neuint.2016.09.004; DOI: 10.1016/j.neulet.2007.05.030; DOI: 10.1523/JNEUROSCI.23-34-10756.2003). We initially referred to it as a “low dose” based on definitions found in previous studies employing similar concentrations (DOI: 10.1186/1750-1326-3-21). However, we fully agree that the magnitude of rotenone’s effects can vary substantially depending on experimental conditions and the specific cell line used. Considering this, we have revised the manuscript to provide a clearer explanation of our dosage selection. In addition, we have removed the reference to “low dose” since, based on the cell viability data obtained in our experimental setting, this concentration cannot be regarded as a low dose capable of inducing a progressive phenotype without acute toxicity.

4. It is necessary to indicate how exactly the cells were grown. On glass or on plastic. (Considering that the materials and methods indicate a confocal microscopy, I assume that the cells were grown on glass coverslips or plates with glass bottom). In what concentration cells were seeded. Were any substrates used to cover the surfaces before seeding cells.

Reply: We apologize to the reviewer for the omission of these details in the previous version of the manuscript. Cells were grown on glass coverslips pre-coated with poly-L-lysine for confocal microscopy, whereas on plastic plates for all the other experiments. According to his/her request, the required details on cell culture conditions are now added in the revised version (lines 529 and 560 of the pdf file).

5. Western blot. Data for antibodies to GAPDH are not provided. SOD-1 protein typically deceted in the range of 18-23 kDa, based on the data from the antibody manufacturer (Santa-Cruz). How do you explain the choice of the lower band, at the level of 15 kDa in images provided (Supplementary page 6). The membranes of the "control" proteins are too overextended in terms of sharpness and contrast. There should be no such noise with the reaction shown by the authors. In case of membrane cutting, please combine the membranes at the cut site so that the entire membrane with the cut line can be seen. Each membrane shows 1 sample, however, the data contains error bars shown in the graphs. Please attach at least 3 membranes to the protein data to understand that the data you obtained are reproducible. SOD-2 beta-Act membrane raises many questions.

Reply: We thank the referee for the careful inspection of the images, here are our detailed comments:

  • The information regarding GAPDH antibody is now reported in Materials and Methods (line 581 of the pdf file).
  • Regarding SOD-1 detection, the upper band around 20–23 kDa visible in the full Western blot representation corresponds to SOD-2, which was analysed on the same membrane. For improved clarity, both bands are now indicated by arrows in the revised figure. SOD-1 typically migrates at approximately 15–18 kDa in its monomeric form (https://doi.org/10.1016/j.redox.2023.102972), with variations depending on species, post-translational modifications, isoforms, and cell lysis conditions. In our experimental conditions, using the Santa Cruz antibody and this specific cell line, only a single immunoreactive band at ~15 kDa was consistently detected.
  • We agree with the reviewer that some Western blots were too overextended in terms of sharpness and contrast, so we decided to repeat some acquisitions for both SOD-1 and SOD-2. Since the original membranes were no longer available, we repeated the experiments using fresh biological samples under identical experimental conditions to improve band resolution and ensure more accurate comparison among treatments. These new blots provided clearer signal definition and a better dynamic range, allowing more precise quantification. Importantly, the overall expression pattern remained consistent with our previous findings. For SOD-2, a sharper and less saturated version of the same membrane has been selected and included as representative in the revised version of the manuscript, as it better reflects the data quantified by densitometric analysis. In the revised version, 3 membranes to the protein data have been added in the Supplementary figures. Wherever possible, the target protein and the loading control are now shown on the same, uncropped membrane.
  • Regarding the SOD-2 beta-Actin membrane, upon re-evaluation of the original blots, we realized that, in the first version of the manuscript, we had inadvertently used an incorrect normalizer for SOD-2. This occurred on the same nitrocellulose membrane where both SOD-1 and SOD-2 were analysed. In that membrane, SOD-1 was correctly shown with the proper housekeeping protein, which has now also been correctly applied to SOD-2 in the revised version. We sincerely apologize for this unintentional oversight and have corrected the figures accordingly. We thank the referee for the careful inspection of the images, which allowed us to identify and rectify this error.

6. The section 4.5, describing how cells we counted and the microscopy of the samples need significant revision. It is necessary to describe exactly how many experiments were conducted. What is meant by the term "experiment". At present, the expression "Morphological analysis was conducted in at least three independent experiments, with a minimum of three images analyzed per experiment.", means, that in the worst case, it is possible to assume that the morphological data are based on 3 wells, from which a total of 9 images were taken (3 images per well). For this type of work, this is simply unacceptable. The count and general condition of the culture must be shown both at low (x10 or x20) and at high magnifications (x40).

Reply: Also following Reviewer 3’s suggestion, dose–response cell counts were carried out, and the overall number of independent experiments was increased to ensure adequate statistical reliability. For each experimental condition, at least three independent experiments (experiments repeated in different weeks) were performed, each comprising triplicate wells, with a minimum of five randomly acquired fields per well analysed. These parameters have now been explicitly stated in the revised figure legend (Figure 1 and 2) and in the Materials and Methods section (Section 4.5).

Furthermore, to improve the representativeness of the morphological analysis, cell counts and quantitative evaluations of both neurite-bearing cells and neurite length were performed on images acquired at 10× magnification, which allows accurate assessment of overall culture density and neurite network organization. In order to better visualize neurite morphology and branching, representative images at 40× magnification are presented in Figure 2. These details have been clearly specified in the Materials and Methods section (Section 4.5) and reflected in the revised figure legend.

7. Microscopy. What are the sizes of cells after differentiation by neuronal phenotype? Check the Scale bar on some images, it may be incorrectly placed or images with incorrect magnification were used (Example Fig. 8. Nat-bLf and Rot + Nat-bLf).

Reply: We would like to clarify that all confocal microscopy images were acquired using the same objective, magnification, and acquisition parameters, and the scale bar of 10 μm was applied to all samples. Therefore, there are no differences in image magnification or scaling between conditions.

The apparent variations in cell size are due to the biological effects of the treatments rather than imaging inconsistencies. In particular, cells exposed to rotenone exhibit marked morphological changes, including neurite retraction and a more rounded cellular shape, consistent with mitochondrial dysfunction and cytoskeletal alterations typically observed in Parkinson’s disease models.

 8. Discussion. The researchers in this paper emphasize that their study is important in the context of Parkinson's disease. However, it would be more correct to say that the results shown cover disorders of all cells susceptible to mitochondrial dysfunction and ROS formation. Considering that rotenone is a lipophilic molecule, it should cause the same effects on undifferentiated cells. This means that if the assumption of Giusi Ianiro and colleagues is correct, Nat-bLf is capable of providing complex protection against ROS in different types of cells.

Reply: we have added this consideration to the discussion section (lines 451-455 of the pdf file).

Reviewer 2 Report

Comments and Suggestions for Authors

This manuscript addresses a critical gap in Parkinson’s disease (PD) research by investigating the differential neuroprotective effects of native (Nat-bLf) and iron-saturated (Holo-bLf) bovine lactoferrin in a rotenone-induced in vitro model of PD.

The finding that Holo-bLf exacerbates oxidative stress and α-synuclein accumulation (even without rotenone) is intriguing but under-explained. The authors speculate that iron saturation drives this phenotype, but direct evidence (e.g., intracellular iron release kinetics from Holo-bLf, or interactions with redox-sensitive signaling pathways like Nrf2/HO-1) is lacking. Additional experiments quantifying labile iron pools or measuring ROS production in situ (e.g., using DCFH-DA) would strengthen the mechanistic link between Holo-bLf’s iron content and pro-oxidant activity.

Author Response

We thank the Reviewer for their helpful comments. The paper has been amended accordingly.

This manuscript addresses a critical gap in Parkinson’s disease (PD) research by investigating the differential neuroprotective effects of native (Nat-bLf) and iron-saturated (Holo-bLf) bovine lactoferrin in a rotenone-induced in vitro model of PD.

The finding that Holo-bLf exacerbates oxidative stress and α-synuclein accumulation (even without rotenone) is intriguing but under-explained. The authors speculate that iron saturation drives this phenotype, but direct evidence (e.g., intracellular iron release kinetics from Holo-bLf, or interactions with redox-sensitive signaling pathways like Nrf2/HO-1) is lacking. Additional experiments quantifying labile iron pools or measuring ROS production in situ (e.g., using DCFH-DA) would strengthen the mechanistic link between Holo-bLf’s iron content and pro-oxidant activity.

Reply: We thank the Reviewer for this insightful comment, which prompted us to further investigate the mechanistic basis of the higher oxidative stress observed in Holo-bLf–treated cells. Following the suggestion, we have now quantified the intracellular labile iron pool, revealing significantly higher levels in cells exposed to Holo-bLf compared to controls. This finding supports the hypothesis that iron saturation contributes to the observed pro-oxidant effect. The results are now described in the main text and showed in the Figure 8.

We also acknowledge the Reviewer’s recommendation to employ direct ROS probes such as DCFH-DA. However, while such probes are useful for detecting short-term ROS fluctuations, their readouts are often influenced by experimental variables such as probe loading, intracellular esterase activity, and differential oxidation by distinct ROS species. These factors can compromise reproducibility and quantitative accuracy, particularly in neuronal cell models. For this reason, we chose to evaluate well-established downstream markers of oxidative stress, including 8-OHdG (oxidative DNA lesions) and 4-HNE (lipid peroxidation). These parameters provide an integrated, time-averaged assessment of oxidative damage, reflecting biologically meaningful outcomes of ROS overproduction rather than transient fluctuations.

Reviewer 3 Report

Comments and Suggestions for Authors

The manuscript reports supporting data for the neuroprotective mechanisms of bovine lactoferrin (bLf) in a cell model of Parkinson’s disease (PD). The authors found that the native form of bLf is more effective than the holo-form in counteracting rotenone-induced cytotoxicity and neurite retraction. This study suggests that the native form of bLf is a therapeutic potential to counteract PD-associated neurodegeneration. However, there are some concerns and suggestions as below.

Major:

  1. The study used 100 mg/mL concentration of bLf in the N1E-115 cells treated with rotenone, and detected the changes of various affected factors such as cell counts. It is strongly suggestive that different amounts of bLf are applied to have a dose-dependent assay especially in Figure 1, so as to obtain more solid and comprehensive data. Plots of a factor (cell counts) v.s. bLf (Nat or Holo) concentration are required in at least some experiments.
  2. Figure 5A, the SOD-1 band (Panel 4) of the rotenone-treated group is obscure in the gel, but an amplitude of 60% appears in the graph as compared with that of the control. It seems that the SOD-1 band (Panel 2) of the Nat-bLf-treated group is much strong, but it is almost equivalent to that of the control or the Holo-bLf-treated in the graph. Analogously, in Figure 7C, the Fpn band (Panel 3) of the Holo-bLf-treated is weaker than that of the Nat-bLf-treated, but it is much larger (stronger) in the graph.
  3. It is reported that α-Syn aggregation is associated with the PD mechanism. However, in Figure 8, only the α-Syn expression (total protein level) is detected and compared by confocal microscopic analysis. It is suggestive that the authors apply a aggregate-bound dye (such as Congo Red) to stain α-Syn aggregates in cells and quantitate the aggregated α-Syn level.

Author Response

We thank the Reviewer for their helpful comments. The paper has been amended accordingly and a point-to-point reply follows.

The manuscript reports supporting data for the neuroprotective mechanisms of bovine lactoferrin (bLf) in a cell model of Parkinson’s disease (PD). The authors found that the native form of bLf is more effective than the holo-form in counteracting rotenone-induced cytotoxicity and neurite retraction. This study suggests that the native form of bLf is a therapeutic potential to counteract PD-associated neurodegeneration. However, there are some concerns and suggestions as below.

Major:

    The study used 100 mg/mL concentration of bLf in the N1E-115 cells treated with rotenone, and detected the changes of various affected factors such as cell counts. It is strongly suggestive that different amounts of bLf are applied to have a dose-dependent assay especially in Figure 1, so as to obtain more solid and comprehensive data. Plots of a factor (cell counts) v.s. bLf (Nat or Holo) concentration are required in at least some experiments.

Reply: We thank the reviewer for this valuable observation, which prompted us to better clarify the rationale behind the selection of the rotenone dosage. Indeed, we initially performed dose-response experiments to identify the most effective concentration. These data were omitted in the previous version of the manuscript but, following the reviewer’s suggestion, they are now included in the revised manuscript to provide a more comprehensive understanding for the reader. Specifically, Figure 1B (cell counts) illustrates the dose-dependent effects of both Nat- and Holo-bLf in rotenone-treated N1E-115 cells.

    Figure 5A, the SOD-1 band (Panel 4) of the rotenone-treated group is obscure in the gel, but an amplitude of 60% appears in the graph as compared with that of the control. It seems that the SOD-1 band (Panel 2) of the Nat-bLf-treated group is much strong, but it is almost equivalent to that of the control or the Holo-bLf-treated in the graph. Analogously, in Figure 7C, the Fpn band (Panel 3) of the Holo-bLf-treated is weaker than that of the Nat-bLf-treated, but it is much larger (stronger) in the graph.

Reply: We thank the Reviewer for highlighting the apparent discrepancies in Fpn and SOD-1 band intensities. The images shown in the figures are representative blots from three independent experiments (now included as supplementary material in the revised manuscript), which explains why the quantified values may not always visually match a single representative blot.

Specifically, in the representative Western blot, the Fpn band in the Holo-bLf–treated sample appears weaker than that in the Nat-bLf–treated sample. However, the corresponding loading control (normalizer) shows a similar relative proportion between these two samples. Since densitometric quantification is normalized to the loading control, the resulting graph can display a slightly higher Fpn signal in the Holo-bLf condition.

Regarding SOD-1, following Reviewer 1’s suggestion to provide clearer and less contrasted western blots, we repeated several experiments for both SOD-1 and SOD-2. The resulting membranes were re-analysed, and the updated data are now included in the revised manuscript. The new representative images better reflect the actual densitometric quantifications, including the higher expression in the sample treated with Nat-bLf alone, improving the visual consistency between the blots and the related graphs.

It is reported that α-Syn aggregation is associated with the PD mechanism. However, in Figure 8, only the α-Syn expression (total protein level) is detected and compared by confocal microscopic analysis. It is suggestive that the authors apply a aggregate-bound dye (such as Congo Red) to stain α-Syn aggregates in cells and quantitate the aggregated α-Syn level.

Reply: We thank the reviewer for the suggestion. However, we do not believe that the use of aggregate-bound dyes such as Congo Red would be useful for the purposes of this work, because alpha-synuclein aggregates or Lewy bodies, being non-amyloid, do not display Congo Red birefringence, which is instead effectively used to visualize amyloid plaques. Additionally, although capable of mimicking the Parkinsonian phenotype not only in animal models but also in cellular models, it has been observed that rotenone (like other molecules employed to induce the Parkinsonian phenotype) is not capable of reproducing Lewy bodies in cellular models (DOI: 10.1186/1750-1326-3-21). Nevertheless, treatment with rotenone in cellular models alters alpha-synuclein homeostasis, leading to its abnormal accumulation, which can be visualized by Western blot or immunofluorescence (DOI: 10.1038/s41419-022-04667-2; DOI: 10.3389/fnmol.2020.560891; DOI: 10.3390/brainsci13040670; DOI: 10.1007/s11064-025-04569-7).

Reviewer 4 Report

Comments and Suggestions for Authors

Great read regarding the possible neuroprotective role of lactoferrin in a Parkinson’s disease in vitro model using rotenone.

Strengths:

    The novelty and originality of the whole in vitro model

    Simple but great experimental design

    Proper statistical analysis

    Good discussion section with well-pointed-out limitations

    High-quality readable language

Limitations:

No major limitations identified, maybe the manuscript might benefit from adding a Conclusions section after the Materials and Methods to clearly summarize the main findings and implications.

Author Response

We thank the Reviewer for their helpful comments. The paper has been amended accordingly and a point-to-point reply follows.

Great read regarding the possible neuroprotective role of lactoferrin in a Parkinson’s disease in vitro model using rotenone.

Strengths:

    The novelty and originality of the whole in vitro model

    Simple but great experimental design

    Proper statistical analysis

    Good discussion section with well-pointed-out limitations

    High-quality readable language

Limitations:

No major limitations identified, maybe the manuscript might benefit from adding a Conclusions section after the Materials and Methods to clearly summarize the main findings and implications.

Reply: Conclusion section was added accordingly.

Reviewer 5 Report

Comments and Suggestions for Authors

Title: Exploring the multifaceted neuroprotective mechanisms of bovine Lactoferrin in a cell culture model of Parkinson’s Disease.

Overall Evaluation

This research article studied the neuroprotective potential of bovine lactoferrin (bLf), in both its Native (Nat-bLf) and Holo form (Holo-bLf), using rotenone-treated N1E-115 cells to mimic PD phenotype. Both forms of bLf preserved tyrosine hydroxylase levels, reduced the DNA damage marker γ-H2Ax, and prevented rotenone-induced downregulation of Divalent Metal Transporter-1 and Ferroprotein, thereby limiting intracellular iron accumulation. The Nat-bLf was more effective than Holo-bLf in counteracting rotenone-induced cytotoxicity and neurite retraction, preserving neuronal morphology and promoting neurogenesis. However, the manuscript is not publishable in the International Journal of Molecular Sciences without major revisions.

Major and minor comments are given below.

  1. -In Figure 1, the authors should compare the Rot and Rot + Holo-bLf treatments to determine if there are significant differences
  2. -In Figure 2, besides the control, the authors should compare Rot, Rot + Nat-bLf, and Rot + Holo-bLf treatments to analyze the neuronal protective effects of Nat-bLf or Holo-bLf in the presence of Rot treatment.
  3. -The images of neurons shown in Figure 2 should be grouped into Figure 2C.
  4. -Figure 4, the TH level seems higher in Rot + Nat-bLf and Rot + Holo-bLf groups than the control. How to explain this observation?
  5. -Figure 8, for the y-axis, what does the % a-synuclein indicate?
  6. -line 85, in vivo; -line 96, line 382, in vitro should be in italic.

Author Response

We thank the Reviewer for their helpful comments. The paper has been amended accordingly and a point-to-point reply follows.

Overall Evaluation

This research article studied the neuroprotective potential of bovine lactoferrin (bLf), in both its Native (Nat-bLf) and Holo form (Holo-bLf), using rotenone-treated N1E-115 cells to mimic PD phenotype. Both forms of bLf preserved tyrosine hydroxylase levels, reduced the DNA damage marker γ-H2Ax, and prevented rotenone-induced downregulation of Divalent Metal Transporter-1 and Ferroprotein, thereby limiting intracellular iron accumulation. The Nat-bLf was more effective than Holo-bLf in counteracting rotenone-induced cytotoxicity and neurite retraction, preserving neuronal morphology and promoting neurogenesis. However, the manuscript is not publishable in the International Journal of Molecular Sciences without major revisions.

Major and minor comments are given below.

    -In Figure 1, the authors should compare the Rot and Rot + Holo-bLf treatments to determine if there are significant differences

Reply: As also suggested by the Reviewer 3, we have now included additional experiments by treating N1E-115 cells with different concentrations of both Nat- and Holo-bLf. The results, now shown in the revised Figure 1B (cell counts), demonstrate the dose-dependent effects of bLf under rotenone treatment. Statistical analysis is now presented between single treatments, when present.

 -In Figure 2, besides the control, the authors should compare Rot, Rot + Nat-bLf, and Rot + Holo-bLf treatments to analyze the neuronal protective effects of Nat-bLf or Holo-bLf in the presence of Rot treatment.

Reply: The statistical analysis shown in Figure 2 already include the comparisons among the indicated treatments. We have re-checked the analysis and confirm that the only statistical difference is between Rot and Rot + Nat-bLf groups.

    -The images of neurons shown in Figure 2 should be grouped into Figure 2C.

Reply: The figure has been amended accordingly.

    -Figure 4, the TH level seems higher in Rot + Nat-bLf and Rot + Holo-bLf groups than the control. How to explain this observation?

Reply: Although TH levels show a tendency to be higher in the experimental groups indicated by the reviewers, this increase is statistically significant only in the Rot + Nat-bLf group. No significant difference was observed in the Rot + Holo-bLf condition compared with the control. Therefore, the absence of statistical significance between these groups indicates that the apparent differences are likely due to random variation rather than true biological effects.. While rotenone treatment typically reduces TH expression through mitochondrial dysfunction and alpha-synuclein-mediated inhibition, Nat-bLf can counteract these effects by both protecting from oxidative damage and reducing alpha-synuclein levels, thereby supporting dopamine biosynthesis and leading to a partial recovery or even increase of TH levels compared with control. This concept is already present in the discussion section (lines 343-345 of the pdf file). We hope that this clarification provides the reviewer with a clearer understanding of the presented results.

    -Figure 8, for the y-axis, what does the % a-synuclein indicate?

Reply: The y-axis in Figure 8 represents the percentage of a-synuclein relative to control condition, as quantified by immunofluorescence intensity. This indication has been added to the y-axis of the Figure.

    -line 85, in vivo; -line 96, line 382, in vitro should be in italic.

Reply: the text has been amended accordingly.

Round 2

Reviewer 3 Report

Comments and Suggestions for Authors

The revised version has addressed the concerns raised, it is now ready for publication.